# Mother-to-Neonate Transmission of Antibiotic-Resistant Bacteria: A Cross-Sectional Study

**DOI:** 10.3390/microorganisms9061245

**Published:** 2021-06-08

**Authors:** Lital Ashtamkar Matok, Maya Azrad, Tamar Leshem, Anan Abuzahya, Thanaa Khamaisi, Tatiana Smolkin, Avi Peretz

**Affiliations:** 1The Azrieli Faculty of Medicine, Bar-Ilan University, Safed 1311502, Israel; litalashtamkar@gmail.com; 2Clinical Microbiology Laboratory, The Baruch Padeh Medical Center, Poriya, Tiberias 1311502, Israel; tleshem@poria.health.gov.il; 3Department of Obstetrics and Gynecology, The Baruch Padeh Medical Center, Poriya, Tiberias 1311502, Israel; ananabuzahya@gmail.com (A.A.); Thm.khm@hotmail.com (T.K.); 4Department of Neonatology and Neonatal Intensive Care Unit, The Baruch Padeh Medical Center Poriya, Tiberias 1311502, Israel; TSMOLKIN@poria.health.gov.il

**Keywords:** antibiotic resistant bacteria, colonization prevalence, carbapeneme-resistant *Enterobacterales*, extended spectrum β-lactam-producing *Enterobacterales*, mother-to-neonate transmission, methicillin-resistant *Staphylococcus aureus*, vancomycin-resistant *Enterococci*

## Abstract

We evaluated carriage rates of extended spectrum β-lactam-producing *Enterobacterales* (ESBL-E), Carbapeneme-resistant *Enterobacterales* (CRE), vancomycin-resistant *Enterococci* (VRE), and methicillin-resistant *Staphylococcus aureus* (MRSA) among pregnant women and determined the maternal-to-neonate transmission rates of these antibiotic-resistant bacteria (ARB). Pregnant women provided rectal and vaginal samples, proximal to delivery. Stool samples were collected from newborns within 48 h of birth. All samples were cultured on selective media for ARB identification. Clinical and demographic data were collected from the participants’ medical files. We performed molecular and phenotypic characterization of the different resistance mechanisms, and determined the isolates’ antibiotic susceptibility and biofilm-forming ability. The prevalence of ESBL-E, MRSA and VRE among pregnant women were 16%, 6% and 1%, respectively. The prevalence of ESBL-E and MRSA among neonates were 7.6% and 1.6%, respectively. Maternal-to-neonate transmission rates of ESBL-E and MRSA were 48% and 27.8%, respectively. Maternal and neonatal isolates shared similar characteristics. Maternal-to-neonate transmission of ARB plays an important role in bacterial colonization in newborns. Future studies should investigate the outcomes of the high ESBL-E transmission rate. The biofilm-forming ability of ARB was found to affect transmission. Additional factors should be investigated in order to understand the differences between transmitted and non-transmitted bacteria.

## 1. Introduction

In recent years, there has been an increased interest in the carriage rates of antibiotic-resistant bacteria (ARB) in the healthy population, with a focus on understanding the impact of long-term carriage and identifying risk factors for carriage and the development of future infection [1]. One population studied in this context is pregnant women. Pregnant women usually comprise a healthy population, but physiological changes during pregnancy (such as cervical enlargement, uterine growth and stress on the bladder) may lead to morbidity [2].

One of the main interests in the ARB carriage rate among pregnant women derives from the possibility of transmitting these bacteria to the neonate during vaginal birth, which may result in severe neonatal infections [3,4,5,6].

Around 23% of the 2.9 million deaths within the first 28 days of newborns’ lives every year were attributed to infectious diseases, with neonatal sepsis accounting for 15% [7]. In recent years, the prevalence of ARB causing early- and late-onset sepsis in neonates has increased [8,9].

Additionally, these bacteria are one of main agents of outbreaks in the neonatal intensive care unit (NICU) [10,11,12]. Neonatal infections with ARB (such as ESBL-producing *E. coli* and MRSA) were associated with increased morbidity and mortality [13,14]. Notably, the rising prevalence of ARB infections is accompanied by increased colonization rates both in the community and in medical institutions [3,6,15,16].

Maternal colonization of ARB was identified as a risk factor for colonization in the newborn [2,3,17,18,19,20]. Therefore, studying the prevalence of ARB among pregnant women is very important. Various ESBL colonization rates among pregnant mothers in different countries were reported [3,15,16,17,19,20]. Similarly, the rates of maternal MRSA colonization varied between different geographic areas. In contrast to ESBL and MRSA, data on CPE and VRE prevalence among mothers and neonates are limited.

Since most studies focused on one type of ARB and data regarding this issue in Israel are sparse, we were encouraged to study the prevalence of MRSA, VRE, CRE and ESBL-producing *Enterobacterales* in pregnant mothers and their newborns in order to identify risk factors for ARB colonization in mothers and newborns and to determine the rate of vertical transmission of these bacteria. Additionally, we compared maternal and neonatal isolates’ characteristics in order to investigate whether they are similar, and identified factors that may affect transmission to the neonate.

## 2. Materials and Methods

### 2.1. Study Population

The study population included 300 pregnant women aged ≥ 18, who arrived to the delivery room of the Baruch Padeh Medical Center (BPMC), Poriya, Israel and underwent a vaginal delivery, and their 304 newborns. Women delivering and babies born via a Caesarean surgery were excluded from the study.

The study was approved by the BPMC Helsinki ethics committee (approval No. POR-0099-18). Each pregnant woman signed a consent form before enrollment. For the newborn’s enrollment in the study, both the mother and the father provided a signed consent.

### 2.2. Sample Collection

Rectal and vaginal samples were collected from pregnant women (mothers) proximal to delivery using cotton swabs (Amies swab, COPAN Diagnostics, CA, USA) and were immediately transferred to the laboratory. Stool samples were collected from the neonates during the first 48 h following birth. Maternal and neonatal samples were first seeded on four different selective chromogenic agar plates—CHROMID ESBL agar (bioMérieux, Durham, NC, USA), CHROMID MRSA agar (bioMérieux, Durham, NC, USA), CHROMagar VRE (HY-Labs, Rehovot, Israel) and CHROMagar mSuperCARBA (HY-Labs, Rehovot, Israel), which were incubated for 24–48 h at 37 °C under aerobic conditions (GasPakTM EZ, BD, USA).

Clinical and demographic data were collected from all participants’ medical records.

A six-month follow-up was performed via a phone call once every 3 months, in which mothers were asked to report if an infection occurred among the newborns carrying antibiotic-resistant bacteria.

### 2.3. Bacterial Isolation and Identification

Following incubation, agar plates were screened for bacterial growth. Following that, bacterial identification was performed by colony morphology (color and size). Final identification was carried out using matrix-assisted laser desorption ionization-time of flight (MALDI-TOF) technology (Bruker Daltonics, Bremen, Germany).

### 2.4. Antibiotic Susceptibility Testing

The Disk diffusion method (Kirby Bauer) was used in accordance with the European Committee on Antimicrobial Susceptibility Testing (EUCAST) 2019 guidelines to determine the antimicrobial susceptibility of ESBL-producing bacteria and MRSA. Isolated colonies were suspended in 0.85% saline in order to create a 0.5 McFarland standard.

For ESBL isolates, two sets of antibiotics were tested. The first set was applied to confirm ESBL production (cefotaxime/clavulanic acid, cefotaxime, ceftazidime/clavulanic acid and ceftazidime (BD Diagnostics, Sparks, MD, USA)). The second set included antibiotics commonly used to treat infection caused by ESBL-producing bacteria (amoxicillin/clavulanic acid, amikacin, gentamycin, ciprofloxacin, trimethoprim-sulfamethoxazole, fosfomycin, nitrofurantoin, ertapenem, meropenem and chloramphenicol (BD Diagnostics, Sparks, MD, USA)).

To assess MRSA, the following antibiotics that are commonly used to treat infection caused by MRSA were tested: clindamycin, fusidic acid, trimethoprim-sulfamethoxazole, ciprofloxacin, erythromycin, mupirocin and gentamycin (BD Diagnostics, Sparks, MD, USA).

Susceptibility to vancomycin was determined using the Etest method, which quantifies the minimum inhibitory concentration (MIC) (i.e., the minimal concentration (μg/mL) of a given antibiotic that inhibits the growth of a particular bacterium under specific experimental conditions). An Etest strip (bioMérieux, Durham, NC, USA) consists of a predefined gradient antibiotic concentration. Isolated colonies are suspended in saline 0.85%, to create a 0.5 McFarland standard. This suspension is seeded on Mueller–Hinton agar with sheep blood (MH blood agar) (HY-Labs, Rehovot, Israel). Subsequently, a vancomycin strip is added to each agar plate and the agar is incubated for 24 h at 37 °C under aerobic conditions (GasPakTM EZ, BD, USA).

### 2.5. Characterization of ^bla^ESBL Genes

All isolates were examined for the presence of the ^bla^CTX-M gene and its allele group, using a multiplex polymerase chain reaction (PCR), as previously described [21]. Isolates found to be ^bla^ _CTX_-_M_ PCR-negative were further examined for ^bla^ ESBL of the ^bla^
_TEM_ and ^bla^
_SHV_ groups.

### 2.6. ESBL Typing

Molecular typing for ESBL-producing bacteria was performed either by repetitive extragenic palindromic (REP)-PCR for *E. coli* isolates or BOX-PCR for *K. pneumoniae*, as previously described [22].

### 2.7. Detection of pvl, mecA and mecC Genes in MRSA Isolates

The presence of Panton–Valentine leucocidin (pvl) was determined in a rapid isothermal amplification reaction, which was performed using the eazyplex MRSAplus kit (AmplexDiagnostics GmbH, Werkstrasse, Germany). This qualitative molecular test detects the presence of the pvl toxin gene and identifies MRSA isolates by the presence of mecA and mecC genes. When mecA, mecC or pvl genes are present in the detected *S. aureus*, specific amplification products are generated. Due to the binding of fluorescence dye to the double-stranded DNA amplification products, the presence of the corresponding genes is visualized via real-time fluorescence.

### 2.8. Detection of vanA and vanB Genes in VRE Isolates

The Xpert vanA/vanB PCR assay (Cepheid, Sunnyvale, CA, USA) was used to detect vanA and vanB genes in suspected VRE bacteria. For this purpose, several colonies were suspended in the sample reagent provided with the kit and then transferred to the test cartridge, which was then loaded onto the GeneXpert instrument. Results were interpreted according to the manufacturer’s instructions, and were categorized as detected/not detected.

### 2.9. Detection of CRE Mechanism

The Xpert CARBA-R PCR assay (Cepheid, Sunnyvale, CA, USA) was used to detect CRE components (KPC, VIM, OXA, IMP and NDM) in suspected CRE bacteria. For this purpose, several colonies were suspended in the sample reagent provided with the kit and then transferred to the test cartridge, which was then loaded onto the GeneXpert instrument. Results were interpreted according to manufacturer’s instruction and were categorized as detected/not detected.

### 2.10. Detection of Biofilm Formation

The ability to form a biofilm was detected using the Congo Red agar (CRA) method. Isolates were cultivated in Congo Red agar that contained brain heart infusion broth 37 g/L, sucrose 50 g/L, agar No. 1 (Thermo Fisher Scientific Oxoid Ltd., Basingstoke, UK) 10 g/L and Congo Red indicator 8 g/L (Thermo Fisher Scientific Oxoid Ltd., Basingstoke, UK). The agar plates were incubated at 37 °C for 24 h under aerobic conditions. The identification of slime-forming strains was done according to colonies’ colors, as previously described [23,24]—biofilm producers appeared as black colonies with a dry crystalline consistency on the red agar, non-biofilm producers appeared as red-colored colonies and black colonies with the absence of a dry crystalline colonial morphology indicated a moderately positive biofilm production.

### 2.11. Statistical Analysis

Numerical data was tested for normality and presented as mean ± standard deviation. Categorical data were presented as frequencies and percent.

Pearson’s Chi Squared Test and Linear model ANOVA were performed in order to compare the distribution of categorical and continues variables, respectively, between two groups (for example, between colonized and non-colonized mothers).

Univariate analysis of odds ratio was performed, with a 95% confidence interval, in order to assess the risk for ARB colonization in neonates that were born to colonized mother compared to a neonate born to a non-colonized mother.

Statistical significance was determined with *p* value < 0.05. All statistical analyses were performed using R statistical language (version 3.6.1, R Core Team 2020).

## 3. Results

### 3.1. Demographic and Baseline Clinical Data of Participating Pregnant Women, Divided to Carrier and Non-Carriers

Out of 300 pregnant women, 64 (21.3%) women were found to be carriers of at least one type of ARB. Five women carries two different bacteria. Table 1 presents the demographic and clinical data of the study group, divided according to carriage status. No significant statistical differences were found in the demographic characteristics (age, nationality and settlement type) of the two groups.

There were no significant differences in the clinical data of both groups, except for the mode of membrane rupture; more women in the carrier group underwent an artificial rupture of membrane compared to the non-carriers (65.6%, 46.6%, respectively, *p* value = 0.007) (Table 1).

### 3.2. Prevalence of Antibiotic Resistant Bacteria among Mothers and Neonates

Sixty-four women were carrier of ARB, with five women carrying more than one ARB type. Therefore, the prevalence of ARB among pregnant women was 23% (69/300). Particularly, 48 ESBL-producing *Enterobacterales* (38 *E. coli* and 10 *K. pneumoniae*) were isolates; 18 and 3 women carried MRSA and VRE, respectively. Therefore, prevalence of ESBL, MRSA and VRE among pregnant women were 16%, 6% and 1%, respectively.

Three-hundred and four (136 females and 168 males) newborns were born to the participating mothers, including four pairs of twins. Twenty-six neonates were found as carriers, with two neonates carrying two types of ARB. Therefore, the prevalence of ARB among the neonates was 9.2% (28/304). In particular, 23 neonates carried ESBL-producing *Enterobacterales*, and 5 carried MRSA, resulting in ESBL and MRSA colonization prevalences of 7.6% and 1.6%, respectively.

It should be mentioned that none of the carrier neonates developed an infection during the 6 months of surveillance.

### 3.3. Comparison of Demographic and Clinical Data of All Participating Neonates Divided According to Carriage Status

Two hundred and thirty-nine neonates were born to 236 non-carrier mothers (three pairs of twins), and 65 newborns were born to the 64 carrier mothers (one pair of twins). Twenty-six (8.55%) neonates, all born to a carrier mother, were found to be carriers to resistant bacteria. None of the neonates that were born to a non-carrier mother were found to be carriers of ARB.

No significant differences were found between carrier and non-carrier neonates in birth weight, gender, antibiotic use during pregnancy, gravidity, delivery type or gestational week (Table 2). In contrast, the two groups differ in the mode of membrane rupture; more neonates from the carrier group were born to a mother who underwent an artificial membrane rupture, compared to the non-carrier group (76.9% vs. 48.6%, respectively) (*p* = 0.006).

Another statistically significant difference between these groups was antibiotic use during delivery; the number of neonates born to mothers who were treated with antibiotics during delivery was higher in the carrier group compared to the non-carrier group (42.3% vs. 19.5%, respectively) (*p* = 0.007).

### 3.4. Comparison of Demographic and Clinical Data of Neonates Born to a Carrier Mother, Divided According to Carriage Status

Out of 65 neonates to carrier mothers, 26 (40%) were found carriers of ARB (Table 3). There were no statistical significant differences between carrier (*N* = 26) and non-carrier (*N* = 39) neonates in birth weight, gender, gravidity, antibiotic use during pregnancy, delivery type, gestational week or amniotic fluid port.

Higher percent of mothers to a carrier neonate have taken antibiotics during delivery, compared to mothers of non-carrier neonates (42.3% and 21.7%, respectively) (*p* = 0.032 (Table 3).

### 3.5. Resistance Mechanisms and Bacterial Type of Bacterial Isolates

One of our aims was to compare the maternal and neonatal isolates’ characteristics in order to investigate whether they were similar and to compare transmitted and non-transmitted bacteria in order to identify factors that may affect transmission to the neonate.

The most prevalent resistance mechanism among maternal and neonatal isolates was ESBL, and the most common bacterium was *E. coli* (Figure 1).

No significant statistical differences were seen between transmitted and non-transmitted bacteria (Table 4).

### 3.6. Bacterial Characteristics of ESBL-Positive Isolates

Out of the 48 maternal ESBL-producing isolates, 23 (48%) were apparently transmitted to the neonates (TN). Most of the isolates among the group of non-transmitted isolates (NTN) and the TN group were *E. coli* (80% and 78.3%, respectively) (Table 5).

In the NTN group, all isolates had the CTX-M gene; 21 isolates of the TN group were positive to CTX-M and 2 (*K. pneumoniae*) were positive to SHV.

The colonies’ number was significantly different between the groups: while most (74%) TN isolates had more than 10 colonies, while most (40%) NTN isolates had less than 5 colonies (*p* < 0.001).

#### Biofilm Formation in Transmitted ESBL Isolates Compared to Non-Transmitted Isolates

Another significant difference between TN and NTN ESBL isolates was seen in biofilm formation (*p* < 0.001); most (92%) of the NTN isolates did not form a biofilm. In contrast, most (56.6%) of the TN isolates were not only biofilm producers, but they were also strong biofilm producers (Table 5).

It should be noted that all neonatal isolates’ characteristics were quite similar to the characteristics of the maternal TN isolates.

### 3.7. Bacterial Characteristics of MRSA Isolates

Out of 18 maternal MRSA isolates, 5 (27.7%) were apparently transmitted to the neonates (TN) (Table 6). All 18 isolates were positive to mecA and negative to mecC. The pvl toxin gene was found only in the NTN group (30.8%). In both groups, most isolates had more than 10 colonies.

#### Biofilm Formation in Transmitted MRSA Isolates Compared to Non-Transmitted Isolates

The TN and NTN isolates differed in biofilm formation; while most (92.3%) NTN isolates did not create biofilm at all, most (60%) TN isolates were strong biofilm producers (Table 6).

As with the ESBL isolates, the MRSA neonatal isolates’ characteristics resembled the maternal TN isolates’ characteristics.

### 3.8. Bacterial Characteristics of VRE Isolates

*Enterococcus faecium* displaying resistance to vancomycin were isolated from three pregnant women, none of whom transmitted the bacteria to their newborn. All three isolates were positive for the VanA gene and negative for the VanB gene (data not shown).

### 3.9. Antibiotic Susceptibility of the Isolates

Our next aim was to compare between the antibiotic susceptibility of the isolates that were transmitted to newborns and the isolates that were not transmitted, as well as between maternal and their neonates’ isolates. Among the ESBL isolates, no significant differences were seen in the antibiotic susceptibility profiles of the isolates from the TN and the NTN groups (Figure 2A). The susceptibility profiles of the neonatal isolates were identical to those of the maternal TN isolates.

Among the MRSA isolates, there were some differences between the TN and the NTN groups (Figure 2B); for example, higher susceptibility rates to clindamycin and erythromycin were seen among the NTN group, as compared to the rates seen in the TN group (53.8% vs. 40% and 53.8% vs. 20%, respectively). Another example is the susceptibility rate to ciprofloxacin, which was 30.8% among the NTN isolates, compared to zero among the TN isolates. In contrast, all TN isolates were resistant to ciprofloxacin. However, no statistical analysis can be performed due to the low number of isolates.

Similar to ESBL, the susceptibility profiles of the MRSA neonatal isolates were identical to those of the maternal MRSA TN isolates.

### 3.10. Maternal-to-Neonate Transmission

As mentioned above, 28 bacterial isolates apparently transmitted from mothers to neonates, resulting in ARB transmission rate of 40.6% (Figure 3). Specifically, the transmission rates of ESBL and MRSA were 48% (23/48) (Figure 3A) and 27.8%, respectively. The odds ratio (OR) of bacterial colonization in newborns born to a carrier mother was 321 (95% CI: 19-5381, *p* < 0.001) as compared to newborn born to a non-carrier woman. In particular, the OR of ESBL colonization in newborn born to a mother carrying ESBL-producing bacteria was 454 (95% CI: 27-7714, *p* < 0.001), compared to a newborn born to a non-ESBL carrier mother. The OR of MRSA colonization in a newborn is 233 (95% CI: 12-4442, *p* < 0.001) among newborns born to women with MRSA carriage compared to women with no MRSA.

## 4. Discussion

In the current study, the prevalence of ESBL, MRSA and VRE colonization in mothers were 16%, 6% and 1%, respectively. A previous study conducted in south Israel found that 21.5% of mothers to preterm neonates were colonized with ESBL-producing bacteria, a prevalence that resembles our result [6]. Other studies have reported on various colonization rates, ranging from 0.1% to 15% [3,15,16,17,18,19,20]. As the colonization rates differ between different countries, some researchers believe they reflect ESBL prevalence among the general community in each country [16,18]. Indeed, a previous study demonstrated that African or Asian origin was a risk factor for ESBL colonization in pregnant women, and suggested that this was a result of the higher carriage rates in these geographical areas [16].

However, as no screening tests for ESBL colonization are performed in Israel, it cannot be concluded whether the colonization rate in our study resembles the general population’s colonization rate.

No CRE was detected in the current study. To best of knowledge, only a few studies have evaluated CRE presence among pregnant women. One study, conducted in Brooklyn, found CPE in 2% of the 100 pregnant women who were screened [25]. In a recent study from Algeria, 4.6% of the participating women carried OXA-48-producing *Enterobacterales* [26]. Since carbapenem antimicrobials are the last source of treatment for MDR bacteria, we assume that their use among pregnant women is probably limited, thus explaining our result.

More studies should be performed in order to determine the colonization rates of CRE among mothers.

The maternal MRSA colonization rate was higher compared to previous reports [3,27,28,29,30,31,32]; for example, Denkel et al. [3] found MRSA in 0.6% of mothers to very low birth weight neonates. In contrast, a higher (29.6%) prevalence of *S. aureus* colonization was found in Africa [27]. The authors suggested that the high prevalence can be attributed to shared breastfeeding, crowded living conditions, high temperatures and humidity [27]. Thus, similarly to ESBL prevalence, MRSA prevalence in pregnant women also varies in different geographic areas.

Interestingly, it was found that *Streptococcus agalactiae* (GBS) colonization increased *S. aureus* rectovaginal colonization [32]. Unfortunately, no evaluation for GBS presence was performed in the current study. Future study should be performed in order to test whether GBS colonization is indeed a risk factor for *S. aureus* colonization and whether the GBS carriage rate in pregnant women in Israel correlates with the MRSA carriage rate.

Regarding VRE colonization in pregnant women, there is only sparse data in the literature. Miller et al. [33] have found that 5.6% of the screened pregnant women carried *Enterococci* with an intermediate susceptibility to vancomycin. However, none of these isolates carried a resistance gene (Van A/B). In a study from Iran, no VRE was detected among 602 strains isolated from the vaginal samples of pregnant women [34]. Thus, it seems that VRE prevalence among pregnant women is low, apparently as a result of infection control efforts to reduce vancomycin use in order to prevent the emergence of VRE.

It should be noted that the various studies that investigated ARB colonization in pregnant women differ in the site and the timing of sampling; while some studies sampled the women in their nares, others collected vaginal or recto–vaginal swabs. Additionally, while some studies sampled the pregnant women weeks before birth, our study included women at the delivery room. These differences may contribute to the different reported colonization rates, since it has been already shown that the sampling sensitivity changes according to the sampling area; MRSA colonization rates in pregnant women ranged from 8% in the vagina to 52% in the nares [35]. Additionally, ESBL prevalence has changed with regard to the sampling time; in a study from 2019, ESBL prevalence in delivery day was lower by 58% compared to the first sampling during pregnancy [19].

Another important point is the including criteria for participating women. While we screened all pregnant women that agreed to participated in the study, some studies collected samples only from mothers to newborns that were either born preterm [6,17,30] or with a low birthweight [3]. These criteria may bias studies’ results, which probably do not reflect the prevalence of maternal colonization in all pregnant women.

### 4.1. Risk Factors for Maternal Colonization

An artificial rupture of membrane was associated with colonization. Although there is no support for this evidence in the literature, we assume that artificial rupture of membrane, as well as other external interventions, may contribute to bacterial colonization as this intervention may introduce rectal bacteria into the vagina.

Further research should be performed to allocate additional risk factors for maternal colonization.

### 4.2. Prevalence of ARB Bacteria among Neonates

ESBL colonization prevalence in the current study (7.6%) resembles the colonization rate (6.2%) that was found in a previous study that we conducted in our NICU in 2017 [22].

As with the mothers’ colonization rates, the neonatal colonization rates varied between different studies, from 0.04% to 14.8% [3,6,16,17,18,19,20]. MRSA colonization rates ranged from 0% to 6.1% [3,20,28,29,30,31].

These differences may result from different geographic areas, with diverse infection control programs. Additionally, developing countries (such as Africa) suffer from crowdedness, which might accelerate bacterial transmission rate [36]. The high temperatures and humidity [36] and the local lifestyle costumes (such as shared breastfeeding [37] in Africa) may also contribute to bacterial transmission [27].

No VRE or CRE isolates were detected among the participating neonates. Only a few studies investigated the carriage rates of VRE among neonates; one study did not detected VRE at all [38]. In contrast, Yüce et al. [39] found that VRE was present in 7.3% of the screened neonates from NICU, compared to 0% among healthy neonates. Considering these findings and our result, it is possible that the low/zero prevalence of VRE reflect the efforts of minimizing vancomycin use in the general population so as to prevent the emergence of VRE.

Regarding CRE, to best of our knowledge, only one study has evaluated CRE prevalence in neonates, with 1.6% of the neonates carrying OXA-48-producing *Enterobacterales* [26]. Thus, further studies should be performed to estimate the prevalence of CRE among newborns.

### 4.3. Risk Factors for Neonatal Colonization

In the current study, we found that antibiotic use during delivery and artificial rupture of the membrane were statistically significant higher among the carrier neonates compared to all non-carrier neonates and compared to non-carrier neonates born to a carrier mother. ESBL and MRSA carriage rates among newborns were previously associated with antibiotic use [14,40,41,42]. Antibiotic use during delivery may contribute to the development or survival of resistant bacteria in the pregnant mother that may transmit to the neonate while passing through the birth canal. As explained earlier, artificial rupture of membrane may result in the introduction of rectal bacteria to the vagina that, again, may transmit to the neonate. Future studies should collect more clinical data regarding both the mothers and their neonates in order to reveal more risk factors for colonization.

### 4.4. Resistance Mechanisms and Bacterial Type of Bacterial Isolates

We did not find a statistically significant difference in resistance mechanism and in bacterial type between bacteria that allegedly transmitted to newborns compared to bacteria that did not transmitted. To our best knowledge, this is the first study that simultaneously investigated the carriage rates of ESBL, MRSA and VRE. Most studies focused on one type of ARB. We found only two studies that explored the prevalence of MRSA and ESBL in newborns. Similar to our study, in both studies, ESBL was more common than MRSA, among both newborns and their mothers [3,20]. While *Enterobacterales* are enteric bacteria, *S. aureus* is more prevalent on the skin. Therefore, as we collected recto–vaginal swabs from mothers and stool samples from neonates, it is expected to find higher prevalence of ESBL-*Enterobacterales* compared to *S. aureus*.

Regarding the predominance of *E. coli*, this evidence was also seen in a former study in our NICU [22] and in another study from south Israel [6]. Among the ESBL-producing bacteria, *E. coli* was the most prevalent and *K. pneumoniae* was the second most prevalent. This result may reflect the general distribution of ESBL-producing bacteria. Further studies should be performed in order to test whether there are differences in the distribution of bacterial type and resistance mechanism between bacteria that transmit to neonates and those that do not transmit.

### 4.5. Characteristics of ESBL-Producing Isolates and Comparison between Transmitted Isolates to Non-Transmitted Isolates

When comparing the ESBL-TN and the ESBL-NTN groups, we found two characteristics that were significantly different; first, the colonies number was higher in the TN isolates. This evidence may indicate the larger bacterial load of the TN bacteria that may contribute to transmission. It should be mentioned that the neonatal sampling was performed during the first 48 h of life. A previous study that collected samples from newborns in several times during hospitalization showed that 35.7% of the neonates were colonized with ESBL only in the second or the third sampling [6]. The authors suggested that in the first days of life, neonates have low bacterial density. Another study, performed in our medical center, also indicated a low bacterial load in the first days of life, since most carrier cases were found in the fourth day since birth [22]. Considering this assumption, only bacteria with sufficient bacterial load would grow on the selective agars that we used; thus, TN ESBL isolates may have more CFU compared to non-TN ESBL-bacteria, and this characteristic may contribute to transmission.

### 4.6. Biofilm Formation in Transmitted ESBL-Positive Isolates Compared to Non-Transmitted Isolates

Another significant difference was seen in biofilm formation; most of the NTN isolates did not form a biofilm. In contrast, most of the TN isolates were strong biofilm producers. Biofilm producer bacteria are more difficult to treat, as they have greater adhesion to the human tissues. Therefore, it is possible that the bacteria from the NTN group (which are non-biofilm producers) did not transmit to the neonates, since they were removed from the birth canal before delivery.

To our best knowledge, this is the first study that evaluated the ability of maternal ESBL isolates to form biofilm. In contrast, several studies have investigated this characteristic in *Streptococcus agalactiae* strains. These studies have identified biofilm formation as a promoting factor for vaginal colonization [43,44]. Furthermore, a study by Ho et al. showed that low vaginal pH promoted the biofilm formation of GBS and thus may benefit GBS colonization [45]. Thus, it would be interesting to investigate whether the acid environment enhances biofilm formation in ESBL.

### 4.7. Biofilm Formation in Transmitted MRSA Isolates Compared to Non-Transmitted Isolates

Although no statistical analysis was performed, we may have noticed a difference in biofilm formation ability between TN and NTN MRSA isolates; while most NTN isolates did not create biofilm at all, most (60%) TN isolates were strong biofilm producers. This result strengthens our hypothesis that biofilm formation may contribute to the transmission to neonates.

Deng et al. [46] identified fibrinogen binding adhesins as key factors for S. aureus persistence within the mouse vagina; following the introduction of several S. aureus strain into mice vagina, they recovered S. aureus from the vagina and counted CFU that were significantly lower in mice that were infected with adhesins-deficient strain compared to the Wild-type [46]. Thus, it seems that the biofilm forming ability may affect vaginal colonization.

### 4.8. Antibiotic Susceptibility of the Isolates

Although no statistically significant differences in antibiotic resistance rates were recorded between the transmitted and non-transmitted isolates, it is difficult to conclude whether antibiotic susceptibility affects the transmission status of bacteria, since our sample size of bacteria is very small. Furthermore, no previous study has compared the antibiotic susceptibility profiles of bacterial strains that transmitted from mother to newborn with strain that did not transmit.

Regarding the comparison between antibiotic susceptibility of maternal strains with those of the neonatal strains, it should be noted that the susceptibility rates of neonatal strains were quite similar to those of the maternal TN isolates. This finding support our hypothesis that most newborn’s bacterial colonization result from maternal-to-neonate transmission.

### 4.9. Maternal-to-Neonate Transmission

In the current study, transmission rates of ESBL and MRSA were 48% and 27.8%, respectively. Reported maternal-to-neonate transmission rates of ESBL varied from 14% to 100% [6,15,16,18,19,20]. Transmission rates of MRSA from mother to newborn ranged from 0% to 86.7% [3,17,20,27,28,29,30,31]. These diverse transmission rates probably result from the different study designs, since some of the studies screened specific sub-populations of neonates (such as neonates with a low birth weight or neonates that are hospitalized in NICU). These inclusion criteria may affect the prevalence of ARB carriage and, consequently, their transmission rates. Additionally, transmission rates depend on colonization rates in the mothers, which were already reported to be different in diverse geographic areas, as mentioned above.

Odds ratio analysis has confirmed that maternal colonization of ARB is a risk factor for ARB colonization in neonates. This result is supported by various studies [3,17,18,19,20,27,28].

### 4.10. Clinical Implications

A high vertical transmission rate of ESBL was noticed. This result should be taken into consideration, along with ESBL infections rate in the neonatal units, since today there is no policy of screening neonates and mothers for ARB (and, particularly, for ESBL). Further research should be performed in order to study the outcomes of such a high transmission rate, including a longer surveillance of neonates. Nevertheless, although the transmission rate was high, ESBL colonization prevalence in neonates was low, suggesting that the transmission of ARB is affected by additional factors rather than by maternal colonization.

The results of this study may lead to conclusions that will require amendments in policies regarding management of pregnant women carrying antibiotic-resistant bacteria. Additionally, they may lay the foundation for an empirical antibiotic treatment and monitoring protocols for neonates carrying resistant bacteria and for neonates with suspected infection.

### 4.11. Research Implications

As we found only a few risk factors for colonization of ARB and for vertical transmission, future study should focus on various risk factors for vertical transmission of ARB in order to draw more comprehensive conclusions.

### 4.12. Strengths and Limitations

The study is very innovative, as it was the first to investigate several types of ARB simultaneously. Additionally, no such a comprehensive study was conducted in Israel.

The study has several limitations; first, it was performed in one small medical center in Israel. Thus, the prevalence of ARB and the transmission rate may not represent the epidemiology of ARB in pregnant women and neonates in all Israel. Second, the number of colonized neonates was quite low. Third, some factors that may attribute to neonatal colonization were not followed.

## 5. Conclusions

This study has confirmed that maternal-to-neonate transmission of ARB plays an important role in ARB colonization in newborns. Our results shed a light on the prevalence of ARB among Israeli pregnant women and their newborns.

## Figures and Tables

**Figure 1 microorganisms-09-01245-f001:**
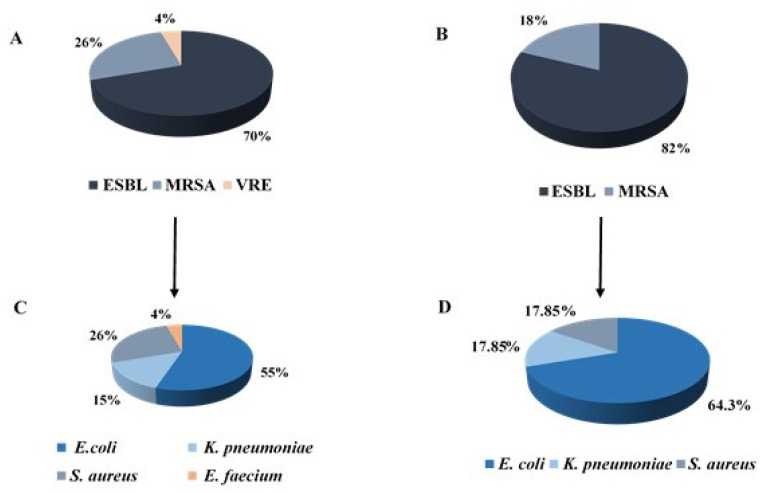
Distribution of resistance mechanisms and bacterial type among all study’s isolates. (**A**,**B**) Distribution of antibiotic resistant bacteria among pregnant women (**A**) and neonates (**B**). (**C**,**D**) Distribution of bacterial species of maternal (**C**) and neonatal (**D**) isolates.

**Figure 2 microorganisms-09-01245-f002:**
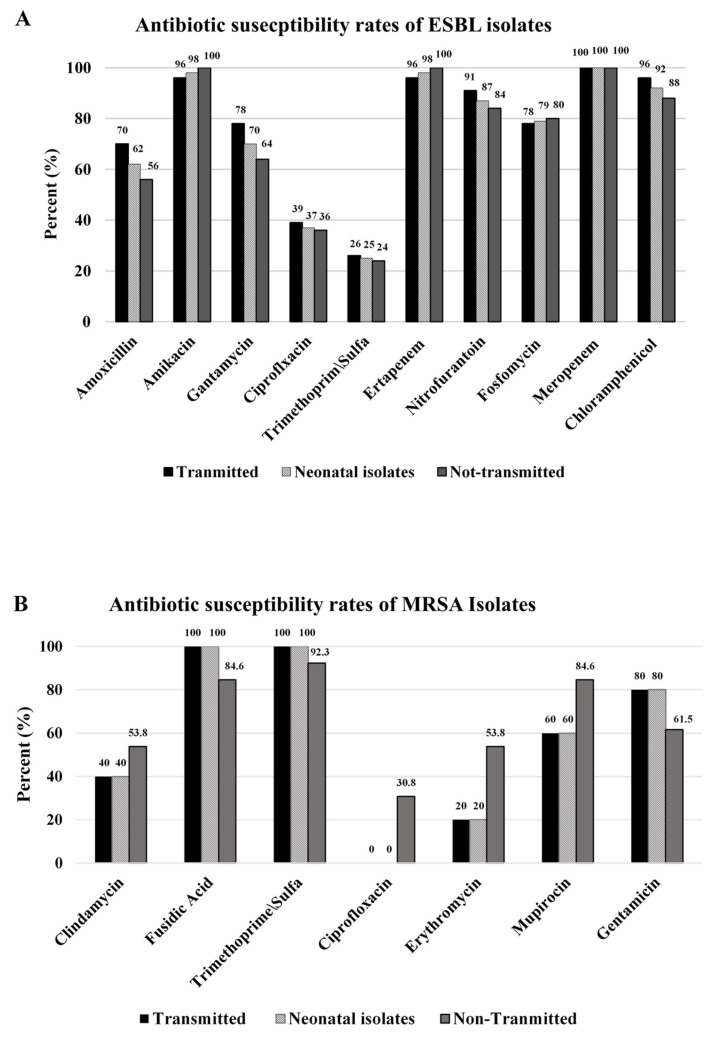
Antibiotic susceptibility rates of ESBL and MRSA isolates. (**A**) Susceptibility profiles of ESBL isolates, divided to maternal transmitted isolates, non-transmitted isolates and neonatal isolates. (**B**) Susceptibility profiles of MRSA isolates, divided to maternal transmitted isolates, non-transmitted isolates and neonatal isolates.

**Figure 3 microorganisms-09-01245-f003:**
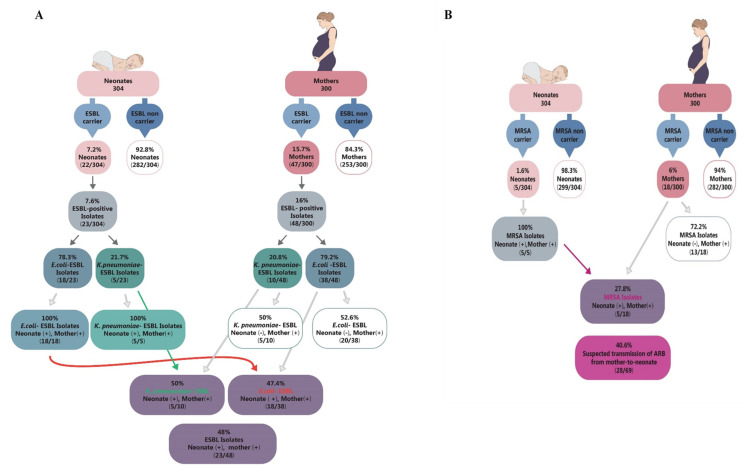
A summary of study’s results regarding maternal and neonatal prevalence of ARB and mother-to-neonate transmission rates. (**A**) Maternal and neonatal carriage rates of ESBL-producing bacteria (divided into *E. coli* and *K. pneumoniae* isolates) and maternal-to-neonate transmission rate of ESBL-producing bacteria. (**B**) Maternal and neonatal carriage rates of MRSA and maternal-to-neonate transmission rate of MRSA.

**Table 1 microorganisms-09-01245-t001:** Demographic and baseline clinical data of participating women.

Characteristic	Non-Carrier (*N* = 236)	Carrier	Total	*p* Value
		(*N* = 64)	(*N* = 300)	
Mean age				
(min–max)	29.5 (18–47)	29.9 (22–42)	29.6 (18–47)	0.602
Nationality (*n*, %)				
Jews	68 (28.8)	25 (39.1)	93 (31)	
Arabs	137 (58)	30 (46.9)	167 (55.7)	0.238
Other	31 (13.2)	9 (14)	40 (13.3)	
Settlement type (*n*, %)				
City	96 (40.7)	32 (50)	128 (42.7)	
Village	2 (0.8)	2 (3.2)	4 (1.3)	0.123
Other	138 (58.5)	30 (46.8)	168 (56)	
Gravidity (Average)	2.17	2.04	2.14	0.517
Antibiotic use during pregnancy				
No	200 (84.7)	60 (93.75)	260 (86.6)	
Yes	36 (15.3)	4 (6.25)	40 (13.3)	0.124
Delivery type				
Vacuum	24 (10.2)	12 (18.75)	36 (12)	
Spontaneous	212 (89.8)	52 (81.25)	264 (88)	0.061
Gestational week (*n*, %)				
≤37	19 (8)	4 (6.3)	23 (7.7)	
38–39	117 (49.6)	33 (51.5)	150 (50)	0.881
≥40	100 (42.4)	27 (42.2)	127 (42.3)	
Amniotic fluid port (*n*, %)				
Spontaneous rupture of membrane (Srm)	126 (53.4)	22 (34.4)	148 (49.3)	
Artificial ruptureof membrane (Arm)	110 (46.6)	42 (65.6)	152 (50.7)	**0.007**
Antibiotic use during delivery				
No	189 (80.1)	47 (73.5)	236 (78.7)	
Yes	47 (19.9)	17 (26.5)	64 (21.3)	0.25

Bold values indicate statistical significance.

**Table 2 microorganisms-09-01245-t002:** Demographic and baseline clinical data of all neonates, divided according to carriage status.

Characteristic	Non-Carrier	Carrier	Total	*p* Value
	(*N* = 278)	(*N* = 26)	(*N* = 304)	
Mean birth weight				
(gr, min–max)	3345.5 (1740–4450)	3282.3 (2180–4380)	3351.4 (1740–4450)	0.436
Gender (*n*, %)				**0.749**
Female	125 (45)	11 (42.3)	136 (44.7)	
Male	153 (55)	15 (57.7)	168 (55.3)	
Gravidity (Average)	2.16	2.11		**0.885**
Antibiotic use during pregnancy (*n*, %)				
No	240 (86.3)	22 (84.6)	262 (86.2)	**0.808**
Yes	38 (13.7)	4 (15.4)	42 (13.8)	
Delivery type (*n*, %)				**0.684**
Vacuum	35 (12.6)	4 (15.4)	39 (12.8)	
Spontaneous	243 (87.4)	22 (84.6)	265 (87.2)	
Gestational week (*n*, %)				**0.376**
≤37	24 (8.7)	1 (3.8)	25 (8.3)	
38–39	140 (50.5)	11 (42.3)	151 (49.8)	
≥40	113 (40.8)	14 (53.8)	127 (41.9)	
Amniotic fluid port (*n*, %)				
Spontaneous rupture of membrane (Srm)	143 (51.4%)	6 (23.1)	149 (49%)	**0.006**
Artificial rupture of membrane (Arm)	135 (48.6%)	20 (76.9)	155 (51%)	
Antibiotic use during delivery (*n*, %)				
No	223 (80.5)	15 (57.7)	238 (78.5)	**0.007**
Yes	54 (19.5)	11 (42.3)	65 (21.5)	

Bold values indicate statistical significance.

**Table 3 microorganisms-09-01245-t003:** Demographic and baseline clinical data of neonates born to a carrier mother, divided according to carriage status.

Characteristic	Non-Carrier	Carrier	Total	*p* Value
	(*N* = 39)	(*N* = 26)	(*N* = 65)	
Mean birth weight	3280.8 (1740–4156)	3282.3 (2180–4380)	3281.4 (1740–4380)	0.99
(gr, min–max)				
Gender (*n*, %)				**0.760**
Female	18 (46.2)	11 (42.3)	29 (44.6)	
Male	21 (53.8)	15 (57.7)	36 (55.4)	
Gravidity (Average)	2.05	2.11	2.07	**0.846**
Antibiotic use during pregnancy (*n*, %)				
No	37 (94.9)	22 (84.6)	59 (90.8)	**0.162**
Yes	2 (5.1)	4 (15.4)	6 (8.2)	
Delivery type (*n*, %)				**0.602**
Vacuum	8 (20.5)	4 (15.4)	12 (18.5)	
Spontaneous	31 (79.5)	22 (84.6)	53 (81.5)	
Gestational week (*n*, %)				**0.220**
≤37	4 (10.3)	1 (3.8)	5 (7.7)	
38–39	22 (56.4)	11 (42.3)	33 (50.7)	
≥40	13 (33.3)	14 (53.8)	27 (41.5)	
Amniotic fluid port (*n*, %)				
Spontaneous rupture of membrane (Srm)	16 (41.1)	6 (23.1)	22 (33.8)	**0.134**
Artificial rupture of membrane (Arm)	23 (58.9)	20 (76.9)	43 (66.2)	
Antibiotic use during delivery (*n*, %)				
No	32 (82)	15 (57.7)	47 (72.3)	**0.032**
Yes	7 (18)	11 (42.3)	18 (21.7)	

Bold values indicate statistical significance.

**Table 4 microorganisms-09-01245-t004:** Characteristics of bacteria, divided according to transmission status.

Characteristic	Non-Transmitted	Transmitted	Total	*p* Value
	(*N* = 41)	(*N* = 28)	(*N* = 69)	
Resistance mechanism				0.114
ESBL	25 (61)	23 (82.1)	48 (69.6)	
MRSA	13 (31.7)	5 (17.9)	18 (26.1)	
VRE	3 (7.3)	0 (0)	3 (4.3)	
Bacterial type				0.225
*E. coli*	20 (48.8)	18 (64.3)	38 (55.1)	
*E. faecium*	3 (7.3)	0 (0)	3 (4.3)	
*K. pneumonia*	5 (12.2)	5 (17.85)	10 (14.5)	
*S. aureus*	13 (31.7)	5 (17.85)	18 (26.1)	

**Table 5 microorganisms-09-01245-t005:** Bacterial characteristics of ESBL-positive isolates.

Characteristic	Maternal Isolates without Transmission to Newborn (NTN)	Maternal Isolates with Transmission to Newborn (TN)	Total	*p* Value	Neonatal Isolates
	(*n*, %), *N* = 25	(*n*, %), *N* = 23	*N* = 48		(*n*, %), *N* = 23
Bacterial type				0.882	
*E. coli*	20 (80)	18 (78.3)	38 (79.2)		18 (78.3)
*K. pneumoniae*	5 (20)	5 (21.7)	10 (20.8)		5 (21.7)
CTX-M presence				0.281	
Yes	25 (100)	21 (91.3)	46 (95.8)		21 (91.3)
No	0 (0)	2 (8.7)	2 (4.2)		2 (9.7)
SHV presence				0.132	
Yes	0 (0)	2 (8.7)	2 (4.2)		2 (9.7)
No	25 (100)	21 (91.3)	46 (95.8)		20 (91.3)
Number of colonies				<**0.001**	
<5	10 (40)	1 (4.3)	11 (22.9)		2 (8.7)
5–10	9 (36)	5 (21.7)	14 (29.2)		5 (21.7)
>10	6 (24)	17 (74)	23 (47.9)		16 (69.6)
Biofilm formation				<**0.001**	
Negative	23 (92)	5 (21.7)	28 (58.3)		6 (26.1)
Moderate	2 (8)	5 (21.7)	7 (14.6)		5 (21.7)
Strong	0	13 (56.6)	13 (27.1)		12 (52.2)

Bold values indicate statistical significance.

**Table 6 microorganisms-09-01245-t006:** Bacterial characteristics of MRSA isolates *.

Characteristic	Maternal Isolates without Transmission to Newborn (NTN)	Maternal Isolates with Transmission to Newborn (TN)	Total	Neonatal Isolates
	(*n*, %), *N* = 13	(*n*, %), *N* = 5	*N* = 18	(*n*, %), *N* = 5
MecA presence				
yes	13 (100)	5 (100)	18 (100)	5 (100)
no	0 (0)	0 (0)	0 (0)	0 (0)
Pvl presence				
Yes	4 (30.8)	0 (0)	4 (22.2)	0 (0)
No	9 (69.2)	5 (100)	14 (77.8)	5 (100)
Number of colonies				
<5	2 (15.4)	0 (4.3)	2 (11.1)	1 (20)
5–10	5 (38.5)	2 (40)	7 (38.9)	3 (60)
>10	6 (46.1)	3 (60)	9 (50)	1 (20)
Biofilm formation				
Negative	12 (92.3)	0 (0)	12 (66.7)	0 (0)
Moderate	1 (7.7)	2 (40)	3 (16.6)	2 (40)
Strong	0 (0)	3 (60)	3 (16.6)	3 (60)

* No statistical analysis was performed due to the low number of isolates.

## Data Availability

The data presented in this study are available on request from the corresponding author.

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
