# Peer review of "Mother-to-Neonate Transmission of Antibiotic-Resistant Bacteria: A Cross-Sectional Study"

_microorganisms, 2021, doi:10.3390/microorganisms9061245_

Round 1

Reviewer 1 Report

In the study "Mother-to-neonate transmission of antibiotic-resistant bacteria: a cross-sectional study", the authors described the medically and epidemiologically important problem of mother-to-child transmission of infection during childbirth.

The study focused on 4 epidemiologically important pathological bacteria - β-lactam - producing Enterobacterales 16 (ESBL-E), carbapenem-resistant Enterobacterales (CRE), vancomycin-resistant Enterococci (VRE), 17 and methicillin-resistant Staphylococcus aureus (MRSA).

An important aspect of the study is also the assessment of the prevalence of pathological bacteria in pregnant women in the Israeli population.

I believe that the weakest part of the paper is the introduction. Therefore, I would recommend that it be improved.

In my opinion, the analytical methods used in the prepared paper are appropriate. 

The authors also used statistical methods appropriate for this type of work. The results of their study are presented in a clear way in the form of 6 tables and 3 figures. Conclusions correspond to the results obtained by the authors. In my opinion, the paper can be accepted for publication after minor corrections.

Author Response

A file is attached.

Reviewer 2 Report

In the manuscript, the authors evaluated carriage rates of important antibiotic-resistant bacteria (ARB) among pregnant women and determined the maternal-to-neonate transmission rates. Through the study, they found that maternal-to-neonate transmission of ARB plays a crucial role in ARB settlement in newborns. I believe this study shows the important mechanism of transmission of ARB from mother to the newborns. However, the following issues are needed to be addressed before it can be published.

  1. In line 182, there are 64 women were found carriers of at least one type of ARB. However, in line 194, there are 69 women were found carriers of ARB. It is confusing. Please check it and revise it.

  1. In line 236, there is a chapter titled 3.5. Bacterial characteristics of study's isolates. However, there are no results according to the section. It could be better to combined chapter 3.5 and 3.6 together.

  1. In figure1B, 19% areMRSA. However, in 1D there are only 15% of S. aureus. It should be matched in two graphs.

  1. In table 5, the percentage of “NO” ofCTX-Mpresence and SHV presence of Maternal isolates with transmission to new-born is not correct. It should be 8.7%. please revise it.

  1. Figure 3 is important to show the transmission process. However, I could not see the text clearly. Please enlarge it for the reader.

Author Response

A file is attached.
